# A New, Validated GC-PICI-MS Method for the Quantification of 32 Lipid Fatty Acids via Base-Catalyzed Transmethylation and the Isotope-Coded Derivatization of Internal Standards

**DOI:** 10.3390/metabo15020104

**Published:** 2025-02-07

**Authors:** Petr Vodrážka, Lucie Řimnáčová, Petra Berková, Jan Vojtíšek, Miroslav Verner, Martin Moos, Petr Šimek

**Affiliations:** 1Biology Centre, Czech Academy of Sciences, Branišovská 31, 370 05 České Budějovice, Czech Republic; vodrazka@bclab.eu (P.V.); rimnacova@bclab.eu (L.Ř.); berkova@bclab.eu (P.B.); 2Department of Chemistry of Natural Compounds, University of Chemistry and Technology, Technická 5, 166 28 Prague, Czech Republic; 3Hospital České Budějovice, B. Němcové 585/54, 370 01 České Budějovice, Czech Republic; vojtisek.jan@nemcb.cz (J.V.); verner.miroslav@nemcb.cz (M.V.)

**Keywords:** fatty acid analysis, human serum, transmethylation, isotope-coded derivatization, GC-MS, positive ion chemical ionization, quantitative analysis, NIST SRM 2378

## Abstract

**Background:** Fatty acids (FAs) represent a ubiquitous class of nonpolar alkyl carboxylate metabolites with diverse biological functions. Nutrition, metabolism, and endogenous and exogenous stress influence the overall FA metabolic status and transport via the bloodstream. FAs esterified in lipids are of particular interest, as they represent promising biomarkers of pathological diseases and nutritional status. **Methods:** Here, we report a validated gas chromatographic-mass spectrometric (GC-MS) method for the quantitative analysis of 32 FAs exclusively bound in esterified lipids. The developed sample preparation protocol comprises three steps using only 5 µL of human serum for Folch extraction, sodium methoxide-catalyzed transesterification in tert-butyl methyl ether, and re-extraction in isooctane prior to a quantitative GC-MS analysis with positive ion chemical ionization (PICI) and selected ion monitoring (SIM). **Results**: The base-catalyzed transmethylation step was studied for 14 lipid classes and was found to be efficient under mild conditions for all major esterified lipids but not for free FAs, lipid amides, or sphingolipids. To minimize matrix effects and instrument bias, internal fatty acid trideuteromethyl esters (D3-FAME) standards were prepared through isotope-coded derivatization with D3-labeled methylchloroformate/methanol medium mixed with each transmethylated serum extract for the assay. The method was validated according to FDA guidelines and evaluated by analyzing NIST SRM 2378 Serum 1 and sera from three healthy donors. **Conclusions:** The measured quantitative FA values are consistent with the reference data of SRM 2378, and they demonstrate the application potential of the described method for general FA analysis in esterified lipids as a novel complementary tool for lipidomics, as well as for the analysis of membrane FAs in dry blood spots and red blood cells.

## 1. Introduction

Fatty acid homeostasis is maintained in all organisms through the circulation and metabolism of a broad spectrum of lipophilic fatty acids (FAs), most of which are esterified in lipids. FAs are essential components of cell membranes, provide energy for ATP formation via the beta-oxidation pathway, and act as starting structures for elongation and desaturation through enzymatic reactions that yield long-chain polyunsaturated FAs (PUFAs). FAs also regulate gene expression, transcription factor activity, intracellular signaling pathways, and the production of bioactive lipid mediators [1,2,3,4]. The increasing knowledge of the role of lipid FAs in human nutrition and various pathological conditions, especially PUFAs (n − 6) and PUFAs (n − 3) [5], requires comprehensive, detailed profiling of individual FAs using efficient, high-resolution separation techniques. Capillary gas chromatography coupled with mass spectrometry (GC-MS) remains a cost-effective tool that meets the latest requirements and is useful in routine diagnostics, as well as in biomedical research, and as a complement to the latest advanced analytical lipidomic strategies [6].

However, most FAs are transported in lipoproteins as phospholipids, triglycerides, and cholesterol esters, and only a small proportion occurs in unbound form. For detailed GC-MS FA analysis in lipids, these bound FAs can first be extracted from the biological matrix, typically through an extraction method commonly used in lipidomics [7], and then the FAs are released via two-step hydrolysis and esterification [8,9] or directly transesterified to FA methyl esters (FAMEs) [10,11]. This step is usually carried out in the presence of an acidic or basic catalyst that is dissolved in methanol under strictly anhydrous conditions. Acid-catalyzed transmethylation/methylation transesterifies not only complex lipids but also free FAs in the presence of methanol [9,11,12,13,14].

Base-catalyzed transmethylation, usually with sodium methoxide in anhydrous methanol, can transesterify lipids directly, and it proceeds much faster. It requires more rigorous anhydrous conditions and is not capable of esterifying free FAs [11,12]. However, the latter has been controversial, as two papers have reported a basic transmethylation of free FAs in biological and food samples [15,16].

FA analysis is usually performed through GC-FID or GC-MS with electron ionization (EI) [6], though negative ion chemical ionization (NICI) has also been reported for ultrasensitive FA detection [8,14]. Positive chemical ionization (PICI) is another established ionization technique in GC-MS with methane and isobutane reagent gases, but it has rarely been used for FA analysis until recently. The application potential of a methane reagent gas for GC-PICI-MS of 37 FAME standards was investigated and compared with the EI mode [17]. Nevertheless, the recent application of a non-standard PICI ion source with acetonitrile as a reagent gas is more promising [18,19].

GC-MS is a well-established tool for quantitative FA analysis, and it is used in most cases with external calibration, usually adding one or more internal FA standards (ISs), which are simultaneously converted to corresponding FAMEs that do not interfere with the target FA analytes [14,20,21,22,23]. Standards for FAs labeled with stable isotopes are now commonly used for this purpose. However, the MS response of analytes in GC-MS analysis is usually influenced by matrix effects during the ionization process and instrument drifts, which are not always efficiently compensated for by the limited number of ISs available for quantitative analysis. To overcome this problem, the technique of the isotope-coded derivatization of regular analytes and their use as internal standards has been introduced into analytical practice [24,25]. The analytes must carry a suitable reactive functional group that can be subjected to the same reaction as the analytes. The standards or control samples are usually labeled with a more expensive, heavy reagent carrying stable isotopes (usually deuterium atoms) and derivatized with the native form of the reagent. Prior to instrumental analysis, the analytes are mixed with native and heavy atoms and subjected to GC-MS or LC-MS analysis [25,26].

The aim of the present work is to investigate four critical aspects of a comprehensive FA analysis in bound lipids: (1) the direct base-catalyzed transesterification of different lipid classes without the esterification of free lipids, (2) isotope-coded calibration for the robust quantification of lipid FAs, and (3) isobutane PICI detection of the investigated FA set and its potential for the quantification of FAME through the isotope-coded preparation of internal standards in serum samples, and (4) method validation according to FDA guidelines in reference NIST SRM 2378 serum samples. Their study paved the way for a novel, validated bioanalytical workflow for the specific analysis of FAs bound in esterified lipids using sodium methoxide-catalyzed transesterification, the improved quantification of a comprehensive set of FAs via isotope-coded internal standard preparation, and robust isobutane GC-PICI-SIM-MS.

## 2. Materials and Methods

### 2.1. Chemicals

Supelco 37 component FAME mix (certified reference material CRM 47885, TraceCERT^®^, in dichloromethane) dissolved in dichloromethane, standards of the 32 corresponding free fatty acids (FFA), methyl heptadecanoate-D33 (MeC17:0-D33), pyridine, methanol (MeOH), methyl chloroformate, (MCF), D3-methanol (D3-MeOH), 12 M hydrochloric acid, tert-butyl methyl ether (MTBE), and chloroform were obtained from Merck (Prague, Czech Republic). Sodium methoxide solution, 25 wt.% in methanol (P/N 156256-25 ML), and potassium methoxide solution, 25 wt.% in methanol (P/N 60402-250ML), were also acquired from Merck, isooctane was purchased from Thermo Fisher Scientific (Pardubice, Czech Republic), and n-Hexane was obtained from Honeywell (Prague, Czech Republic). D3-methyl chloroformate (D3-MCF) was purchased from Cambridge Isotope Laboratories (Cambrige, UK). Deionized water was obtained with a Milli-Q Reference Water Purification System (Molsheim, France).

A set of individual lipids (1-hexadecanoyl-rac-glycerol; 1,2-dihexadecanoyl-rac-glycerol; 1,2,3-trihexadecanoyl-sn-glycerol; 1,2-diheptadecanoyl-sn-glycero-3-phosphate; 1,2-ditetradecanoyl-sn-glycero-3-phosphocholine; 1,2-dihexadecanoyl-sn-glycero-3-phosphoethanolamine; 1,2-di-(9Z-octadecenoyl)-sn-glycero-3-phosphoethanolamine; 1,2-di-(9Z-octadecenoyl)-sn-glycero-3-phospho-(1′-sn-glycerol); 1,2-ditetradecanoyl-sn-glycero-3-phosphoserine; 2-tetradecanoyl-sn-glycero-3-phosphoethanolamine; 1-octadecanoyl-sn-glycero-3-phospho-(1′-sn-glycerol); 1-(9Z-octadecenoyl)-sn-glycero-3-phosphate; 1-(9Z-heptadecenoyl)-glycero-3-phosphoserine; Cholest-5-en-3β-yl octadecanoate; O-hexadecanoyl-R-carnitine; *N*-(heptadecanoyl)-sphing-4-enine-1-phosphocholine; and *N*-(dodecanoyl)-sphing-4-enine and *N*-(dodecanoyl)-1-β-glucosyl-sphing-4-enine) (refer also to Appendix A) were purchased from Avanti Polar Lipids (Alabaster, AL, USA).

### 2.2. Biological Material

The NIST^®^ Standard Reference Material^®^ (https://shop.nist.gov/ccrz__ProductDetails?sku=2378&cclcl=en_US, accessed on 30 December 2024) abbreviated SRM 2378 (NIST store, SKU: 2378; Fatty acids in frozen serum, Serum 1, 2, 3), was used. The sera contain fatty acids with reference concentration values in frozen human serum and are primarily intended for the validation of methods for the determination of total fatty acids in human serum. According to the NIST provider, the first sample was obtained from three healthy donors who had taken 1000 mg/day of fish oil supplements one month prior to collection; the second sample was obtained from three healthy donors who had not taken fish oil or flaxseed oil one month prior to collection [14]. Another pooled human serum from fasting blood samples from anonymous donors for the development of the method and from three healthy adult laboratory participants was collected with written consent according to a routine protocol at the České Budějovice hospital. The study complied with the principles of the Helsinki Declaration and was approved by the local ethics committee for research (České Budějovice Hospital, ethics number LEC 103/20).

### 2.3. Stock and Calibration Solutions

The calibration solutions were prepared by diluting a commercially available stock solution of a certified reference material (CRM 47885) (see Appendix A). An aliquot of the CRM stock solution (90 µL) was evaporated to dryness under a gentle stream and reconstituted in 400 µL of isooctane (concentration level 1). Calibration level 2 was prepared by mixing 200 µL of level 1 solution with 200 µL of isooctane. The same dilution procedure was repeated stepwise to obtain a series of 12 calibration concentration levels.

The prepared free fatty acid standards were dissolved at a concentration of 1 mg/mL using dichloromethane for long-chain FAs (C > 12) and methanol for short-chain FAs. The prepared stock solutions were then diluted in two separate mixtures. The FFA concentrations in these mixtures were carefully adjusted to ensure that the resulting solutions matched the midpoint of the calibration curve after the derivatization step with D3-MCF/D3-MeOH and the addition of internal standards (IS).

### 2.4. Internal Standards

Two calibration methods were investigated in this study.

(1)The established routine GC-EI-MS quantification of FAMEs was performed with a single commercially available internal standard of methyl decanoate labeled with 33 deuterium atoms (MeC17:0-D33) at a concentration of 0.3 mg/mL dissolved in isooctane.(2)For the novel isotope-coded calibration assay, a mixture of trideuterated methyl esters (D3-FAME) was prepared from the free 32-FA standards esterified with trideuterated MCF (D3-MCF) in trideuteromethanol (D3-MeOH) according to the previously described reaction mechanism [27]. The mixture of the isotope-coded D3-FAMEs was prepared manually. The two mixtures of the 16 + 16 free FAs were pipetted into 6 × 50 mm glass tubes (Kimble-Chase, Vineland, NJ, USA) and evaporated to dryness. Then, a mixture of 20 µL of D3-MeOH and 16 µL of pyridine was added. Subsequently, 120 µL of isooctane and 20 µL of D3-MCF were added, and the reaction medium was shaken again. Finally, the reaction was stopped with 100 µL of 1 M HCl, and after vortexing, 100 µL of the upper layer was transferred to a new vial. The procedure was repeated several times to obtain a sufficient amount of internal D3-FAME standards for serial FA analysis. The collected upper isooctane phases were combined and finally aliquoted for the isotope-coded calibration and FAME quantification in specific transmethylated lipid extracts obtained from serum samples.

### 2.5. Individual Lipid Solutions for the Investigation of FA Yields After the Base-Catalyzed Transmethylation Reaction

The individual lipid standards were dissolved in chloroform–methanol (2:1, *v*/*v*). The stock solutions were prepared at a concentration of 1 mg/mL. The final lipid solutions for the determination of FA yields after transmethylation were prepared through the appropriate dilution of the stock solutions of the individual lipids with the same chloroform-MeOH medium (2:1, *v*/*v*).

### 2.6. Extraction of Lipids from Biological Material

Sample extraction was performed using a modified Folch protocol [28]. Briefly, the reference SRM 2378 serum and serum samples used (5 µL each) were pipetted into vials, to which 500 µL of a chloroform–methanol (2:1, *v*/*v*) mixture was then added. The samples were transferred to an ultrasonic bath (10 min) and centrifuged (4 °C/10 min/10,000 RPM). The supernatant was transferred to a new derivatization vial, and the content was evaporated to dryness and used for the subsequent transesterification step.

### 2.7. Base-Catalyzed Transesterification

The optimized protocol involves the addition of 50 µL MTBE to a derivatization vial containing the lipid extract. After vortexing, 100 µL of 2 M MeONa in methanol was added, and the mixture was shaken vigorously for 3 min. The reaction was quenched by adding 150 µL of hexane and 100 µL of 2 M hydrochloric acid. After thorough mixing, the upper hexane layer was carefully collected. To maximize the yield of the FA extraction, another 100 µL of hexane was added to the reaction medium, and the resulting hexane top layer was mixed with the first hexane portion and 10 µL of the corresponding internal standard (IS). MeC17:0-D33 was used as a single internal standard in the traditional protocol, while a mixture of isotope-coded (IC) D3-FAME ISs, prepared separately, was analyzed using the new IC multiple IS protocol. The combined hexane layer containing the corresponding internal standards (ISs) was finally evaporated to dryness using a gentle stream of nitrogen, and the dry extract was reconstituted in 50 µL of isooctane and subjected to GC-MS analysis using either electron ionization (EI) or positive ion chemical ionization (PICI) to detect the FAME metabolites via GC-MS.

### 2.8. GC-EI-MS and GC-PICI-MS Analysis

The separation and quantification of FAMEs was performed using a 7890B gas chromatograph coupled with a 7010B triple quadrupole mass spectrometer (Agilent, Santa Clara, CA, USA) that was equipped with an EI and PICI ionization source. Calibration solutions and transmethylated lipid sample extracts (1 µL in all cases) were injected into a splitless double-cone liner (Agilent, Santa Clara, CA, USA) in splitless mode at constant pressure (valve closed at 0.5 min). The injector temperature was 250 °C, the flow rate of the helium carrier gas (99.999%) was 1.1 mL/min, the temperature of the ion source was 300 °C, and the transfer line was 250 °C. A 30 m × 0.25 mm i.d., 0.20 µm Zebron ZB-FAME GC column (Phenomenex, Torrance, CA, USA) was used. The oven was initially held at 70 °C for 2 min and then increased to 180 °C at 15 °C/min and held for 6 min and increased to 230 °C at 10 °C/min and held for 1 min. The MS detection parameters for the EI mode were as follows: 70 eV, scanning mode, and either full scan, mass range 60–500 Da for the characterization of FAMEs, or Selected Ion Monitoring Mode (SIM) for their quantitative analysis. The Positive Ion Chemical Ionization (PICI) mode applied the following: isobutane reagent gas, scanning mode with either a full scan and a mass range of 60–500 Da or SIM. The Agilent MassHunter Version 10 software was used to process the acquired GC-MS data.

### 2.9. Quantification of the Lipid FAs via Single Internal Standard (Single IS) Calibration and the New Isotope-Coded Multiple Internal Standard (IC-Multi-IS) Calibration Method

Calibration curves, a lower limit of quantification, precision, accuracy, cross-analyte interference, carryover, stability, the matrix effect, and the dilution factor were evaluated in accordance with generally accepted analytical guidelines [29,30]. A twelve-point calibration curve was constructed by plotting the peak area ratio against the corresponding internal standard.

Two calibration methods were tested: (1) the traditional method with an internal standard, in which case 10 µL of MeC17:0-D33 stock solution (total 0.3 mg/mL) was used, and (2) the new IC-multi-IS calibration method, in which a mixture was prepared with corresponding isotope-coded D3-FAMEs. In both cases, a 10 µL aliquot of IS solution was evaporated to dryness in a stream of nitrogen and redissolved with 50 µL of the corresponding calibration solution. The calibration list of working solutions with the FA concentrations used is summarized in Appendix A.

Each calibration point was measured six times. The accuracy and precision of the analytical method were evaluated at the lower limit of quantification (LLOQ), the low QC, the medium QC, and the high QC for 6 replicates. As acceptance criteria, the mean value within ±15% of the nominal value (one exception for the LLOQ ± 20%) was used for precision. Selectivity was evaluated by comparing three types of samples: (a) a blank sample, (b) a blank sample with the internal standards (ISs), and (c) another blank sample with the IS(s) and with FAMEs at the level of the LLOQ. The acceptance criteria were selectivity, CV ≤ 20%, for the LLOQ for each analyte and CV ≤ 15% for the corresponding IS. The interference between analytes was measured for all analytes at their ULOQ level and with the ISs present. The acceptance criteria for the interference are CV ≤ 20% at the LLOQ level for each analyte and CV ≤ 15% for the corresponding IS.

Carryover was checked after the injection of each standard at its ULOQ value by measuring the blank (isooctane, n = 2). The acceptance criteria for each analyte and the corresponding IS response were CV ≤ 20% and CV ≤ 15%, respectively.

The stability of FAMEs in the serum extracts was measured with two calibration levels (low QC and high QC), and the samples were stored at −20 °C for 14 days. The recovery rate was measured in accordance with FDA guidelines [29]. Three concentration levels (low QC, medium QC, and high QC) were measured in pure solvent and in spiked real samples (the pooled human serum), and the recovery rate was calculated. The recovery value must be consistent, accurate, and reproducible; i.e., it must fulfill the conditions for accuracy and precision [29]. The integrity of the dilution was checked using a 1:9 (*v*/*v*) dilution of the FAME with isooctane above the ULOQ value.

## 3. Results

### 3.1. Optimization of the Transesterification Procedure

The base-catalyzed step of converting FAs acylated in lipids to FAMEs was investigated by testing different solvents in the reaction and changing the temperature and reaction time. The observed experimental transesterification data are summarized in Table 1.

In his review, which focused on transmethylation reactions, C.C. Christie generally recommended a wide range of inert solvents for fast base catalysis [31]. Our experiments on the transmethylation of single lipid standards, covering 14 lipid classes, with different inert solvents have shown that the efficiency of the reaction largely depends on the lipid structure, especially its polarity and steric effects. Non-polar solvents such as hexane and toluene are not efficient for the release of FAs from cholesterol esters (CEs), as the yield was below 10% even under harsher conditions (Table 1). The non-polar solvents are not miscible with sodium methoxide, which probably hinders sufficient contact between the non-polar analyte and the reagent during the reaction. Similarly, the experiments with the polar solvent acetonitrile also did not produce satisfactory yields for nonpolar CEs (yields < 10%). However, medium polar tert-butyl methyl ether (MTBE) proved to be a suitable medium for transesterification with sodium and potassium methoxides. In particular, MTBE improved the yield of non-polar cholesteryl esters to satisfactory levels (>80%), Table 1.

Basic catalysis with the sodium methoxide in MTBE proceeds efficiently with various tested lipid class representatives at room temperature within minutes. In comparison to previous methods of methoxide transesterification, we systematically tested commercially available sodium and potassium methoxides (25 wt%) in MeOH and obtained robust results over a period of several months. Reaction times from 1 min to 1 h were tested. Surprisingly, longer reaction times (30 to 60 min) decreased FAME yields for some lipid classes, probably due to the degradation of methyl esters through competitive saponification with residual moisture. Short 1-min reactions were less efficient, especially for non-polar CEs. Higher reaction temperatures (40 °C and 60 °C) also had negative effects on FAME yields.

The use of 2M sodium methoxide with MTBE at room temperature and a reaction time of 3 min proved to be suitable reaction conditions that allowed the robust and efficient transesterification of lipid individuals representing 14 lipid classes with yields above 80% and RSD < 10% (n = 6), Table 1, and without observable side reactions such as cis–trans isomerization [20].

It should be noted that basic reaction conditions form anions from free carboxylic acids, which hinder their esterification [31]. The base-catalyzed method, therefore, does not allow the analysis of free FAs, which must be determined using another method. On the other hand, as we have shown in Table 1 below, transesterification can proceed with other complex esterified/acylated structures such as acylcarnitines, which serve as active FA transporters in the catabolic process of beta-oxidation. We have also demonstrated that methoxide-catalyzed transesterification does not hydrolyze the amide bond in sphingolipid metabolites, as shown by the methoxide reaction in MTBE with sphingomyelin, ceramide, and glucosylceramide (Table 1).

### 3.2. GC-MS Analysis of Lipid Fatty Acids

The combination of the Folch extraction step of the lipid pool with methoxide-based transesterification and calibration with internal standards was investigated for the quantification of 32 FAs. The method provided clean serum extracts and enabled the development of a rapid and robust analytical protocol for the GC-MS quantification of lipid FAs in human serum.

The new method was investigated with a GC-MS system equipped with both EI and PICI ionization, each operated in SIM-MS scan mode. Two internal calibration methods were used to quantify the FAMEs: (i) a single internal C17:0 standard labeled with D33 isotopes and (ii) the new internal multiple standards simultaneously isotopically coded to form the corresponding D3-FAME pair for each particular FAME analysis target.

The standard cyanopropyl GC phase ensures the baseline separation of 32 specific FAMEs, especially cis-, trans-, omega-3, and omega-6 FAMEs, as well as their sensitive and robust detection in SIM-MS mode. The diagnostic ions used for EI-SIM-MS and PICI-SIM-MS detection are listed in Table 2.

PICI-MS with isobutane reagent gas provided unambiguous pseudomolecular [M+H]^+^ ions of all measured FAMEs, including polyunsaturated FAs with minimal fragment ions, which in combination with retention data allows the easy, sensitive monitoring and quantification of all measured FAMEs. In contrast, hard EI ionization is known to generate complex EI mass spectra and the diagnostic ions suitable for quantification must be carefully selected [20].

Characteristic EI and PICI mass spectra of model unsaturated C16:0 and polyenic arachidonic acid (DHA, C20:4(n − 6)) FAs are shown in Figure 1A–F.

While the unsaturated FAMEs such as methyl esters of C16:0 show clear fragment ions in both the EI (Figure 1A for C16:0-Me and Figure 1B for C16:0-MeD3) and PICI mass spectra (Figure 1C for the mixture C16:0-Me and C16:0-MeD3), the situation with the polyunsaturated FAs is complicated, as the examples DHA-Me and DHA-MeD3 show (Figure 1D,E). The complex EI spectra of PUFAs make the identification of suitable diagnostic ions for EI-SIM-MS quantification and even in the case of isotope-coded multiple IS EI-SIM-MS analysis technically infeasible (Figure 1D). On the other hand, the PICI mass spectra of the PUFAs show distinct peaks of pseudomolecular ions for all FAMEs.

### 3.3. FA Calibration and Method Validation

The investigation of suitable calibration strategies, i.e., with a single internal standard and with isotope-coded multiple D3-FAME standards in combination with EI-SIM-MS and CI-SIM-MS detection, resulted in four calibration methods: (A) single IS/EI-SIM-MS, (B) single IS/PICI-SIM-MS, (C) multiple isotope-coded IS/EI-SIM-MS, and (D) isotope-code multiple-IS/PICI-SIM-MS.

All four calibration methods were evaluated for various validation parameters, including linearity, precision, accuracy, selectivity, inter-analyte interference, carryover, stability, RT repeatability, the matrix effect, recovery, and dilution integrity. The validation parameters and calibration data are summarized in Appendix A.

The comparison of the important validation parameters, i.e., the concentration calibration range and the RSD values of the LLOQs, is documented in Table 3.

The summarized analytical results show that satisfactory calibration data and concentration ranges were obtained for all the estimated 32 FAME analytes, except for the polyunsaturated FAs (PUFAs) measured using the IC-multi-IS GC-EI-SIM-MS method, Table 3, for which the suitable diagnostic ions in their EI mass spectra were not feasible for the reasons documented in Figure 1D–E, which illustrate the complex character of the EI mass spectra of PUFAs.

The quantification limit ranges were evaluated in terms of their linear dynamic calibration range (with R2 ≥ 0.985), accuracy (RSD < 20%), and precision (RSD < 20%). Limit calibration data that did not meet the validation criteria were excluded.

The data summarized in Table 3 show the advantages of the new isotope-coded multiple-IS calibration method D (IC-multi-IS) in combination with PICI-MS detection, which outperforms the other compared methods A-C by a wider dynamic range of more than three orders of magnitude and lower limits of quantification (LLOQs) with an average LOQ of 0.045 µg/mL, in contrast to the single-IS EI-MS method with an average LOQ of 0.118 µg/mL.

Based on the data obtained, the final evaluation of the GC-MS methods for FAME quantification in human serum using method A (single-IS GC-EI-SIM-MS) and the new method D (IC-multi-IS GC-PICI-MS) were investigated and compared in detail; see Table 4A,B.

Both Method A and Method D provided satisfactory results. The coefficients of determination (R^2^) estimated via linear regression were ≥0.98, and the LLOQ ranged from 0.022 to 0.18 µg/mL. The precision and accuracy of the method were 0.21–15% and 82–122%, respectively. The FAME derivatives showed only minimal interfering peaks, with the response not exceeding 20% of the LLOQ values, so that the selectivity criteria were met. No cross-analytical interferences were found, so the acceptance criteria were met. No carryover was observed in the concentration areas of interest. The derivatives of the standards proved to be stable under the conditions tested. The repeatability of the retention time was evaluated over several weeks and resulted in a CV of less than 0.05%. The dilution factor test gave satisfactory results for all analytes. The sample dilution of 1:9 can be used if the measured fatty acid concentration exceeds the ULOQ value. This situation can often occur with abundant fatty acids such as oleic acid (C18:1(n − 9)), palmitic acid (C16:0), or stearic acid (C18:0). Finally, the method fulfills the validation criteria in terms of recovery rates.

### 3.4. Quantification of FAMEs in Human Serum via Base-Catalyzed Transmethylation and the Single-IS GC-EI-SIM-MS and IC-Multi-IS GC-PICI-MS Methods

First, the two calibration methods (Table 4A,B) were examined based on the single and multiple IS standard calibrations for the quantitative analysis of lipid FAs as FAMEs in certified NIST 2378 human Serum 1. The results are summarized in Table 5 and were compared to the NIST SRM 2378 reference data. FA concentrations were measured with satisfactory precision (expressed as % CV), and accuracy was assessed by comparing the two calibration methods with t-statistics showing statistically non-significant differences at the 95% confidence level (*p*-value > 0.05).

However, FA values reported in the SRM2378 reference material from Serum 1 were, unsurprisingly, lower, as base-catalyzed transmethylation only allows the analysis of FAs bound in lipids and not FAs present in free form. In contrast, FA concentrations reported for the certified SRM2378 were measured through a more complicated and time-consuming two-step sample preparation involving base hydrolysis, followed by acid esterification [7,14], and they refer to the analysis of all FAs in the serum sample.

GC-PICI-MS analysis using the new, more sensitive IC-multiple-IS GC-PICI-SIM-MS method showed consistently lower values for all abundant FAs than the reference SRM2378 Serum 1. The change was quantified using fold-change (FC) values, and it ranged mostly from −10 to −27%.

Fold-change (FC) values were further calculated for the summed FA concentrations in specific FA families, labeled ∑ SAFA, ∑ MUFA, ∑ PUFA (n − 6), and ∑ PUFA (n − 3) and ∑FA in Table 5. The result data indirectly show that about 11% of saturated, 17% of ∑MUFA, 23% of ∑PUFA (n − 6), 21% of ∑ PUFA (n − 3) and, in summary, about 18% of abundant FAs can circulate as free FAs in human serum. In contrast, a comparison of the commonly used index of the ratio of PUFA (n − 6/n − 3) [32,33] indicates close values with a slightly higher abundance of PUFAs bound in serum lipids.

The new IC-multi-IS GC-PICI-MS method was finally applied for FA analysis in sera from three volunteers with ages of (42, 41, and 64 years) and BMIs of (23.7, 27.7, and 27.8). The healthy patients from Central Europe had a varied diet with chicken, pork, and beef but consumed less fish and did not take PUFA supplements. The resulting FA concentrations are summarized in Table 6 and compared with the SRM 2378 serum levels of three subjects who did not receive PUFA supplementation.

A GC-MS analysis of FAs bound in the lipids provided consistent data on the most abundant lipid FAs observed in similar amounts. However, the profile of total FAs in SRM 2378 Serum 3 showed significant differences; in particular, the tabulated total Σ PUFA a (n − 3) abundance was significantly lower in SRM 2378 Serum 3 (Western US diet) than in the sera from donors with Central European dietary habits (Table 6, FC = 47.0%).

Regardless of the comparison of the measured lipid-bound FAs with the total FA content in SRM 2378 sera, the fold-change values of Σ saturated FAs remained at the same level (both FC = −11%) and correspond to the approximate 11% proportion of free FAs in the measured and reference serum samples (Table 5 and Table 6).

The summed abundance of the major FA families in the measured serum samples and the reference values of SRM 2378 Serum 1 and Serum 3 are shown in Figure 2.

The distribution of FA metabolites among the major FA families in Figure 2 provides a clear picture of the Western diet of the serum donors studied, where PUFA (n − 6) intake dominates and PUFA (n − 3) is lowest. This feature is characteristic of the SRM 2378 reference samples Serum 1 and even more of Serum 3, presumably from donors from the USA with Western habits and a diet rich in seed and vegetable oils and processed foods with a low proportion of fish meal. Nevertheless, the patient’s FA supplementation withfish oil as reported in SRM 2378 Serum 1 apparently lowered the ∑ PUFA (n − 6)/(n − 3) ratio from an undesirably high value of 12.7 (Table 6) to an acceptable ratio of 5.6 (Table 5).

The lipid FA distribution in the sera of Central European donors measured using the developed GC-PICI-SIM-MS method differs (Table 6) even when the lipid FA levels are compared with the total FAs declared in the SRM 2378 reference samples. Nevertheless, the ∑ PUFA (n − 6)/(n − 3) ratio in the sera of the Central European donors remains high at about 9 (Table 6), and as shown in Table 5, regular PUFA (n − 3) supplementation with fish oil may efficiently decrease the ratio so that it is closer to the recommended values (<4) [5,32]. Although the relative content of the ∑ PUFA (n − 3) was found to be 50% higher than in the reference SRM 2378 Serum 3, its relative proportion to the whole FA pool in serum was still found to be low at about 3–6%, even after PUFA supplementation (Table 5 and Table 6).

## 4. Discussion

We tested four GC-MS methods for the investigation of fatty acid metabolism via the quantification of lipid-bound FAs in human serum; each included one of two calibration procedures (single-IS or IC-multi-IS) and one of two GC-SIM-MS methods using either the EI or PICI ionization modes. The results are summarized in Table 7.

The combination of the new IC-multi-IS calibration with GC-PICI-SIM-MS analysis was found to be robust and sensitive, with the largest calibration range of more than three orders of magnitude. The entire analytical workflow is summarized in Figure 3.

The new GCMS method requires simple, efficient sample preparation in three steps with only 5 µL of human serum for Folch extraction, sodium methoxide transesterification in methanol-TBME (2:1), and re-extraction in isooctane prior to quantitative GC-PICI-SIM-MS analysis. The base-catalyzed transmethylation step was investigated in detail using 14 lipid classes. Commercial sodium methoxide in MeOH (25 wt%) with the solvent MTBE proved to be the most efficient medium for the release of FAs bound in esterified lipids including cholesterol esters, but not in lipid amides, under mild conditions at room temperature in about three minutes. Free FAs and sphingolipids are not transesterified and are not available with the described method.

All 32 FAMEs investigated yield distinct [M+H]^+^ ions, which are suitable for the direct identification of specific GC-resolved FAMEs by assigning their retention times and simple PICI mass spectra. The work shows that the use of the PI-CI ionization mode for the global quantitative GC-MS analysis of FAMEs has been underestimated but represents a perspective analytical strategy using either isobutane or non-standardized acetonitrile as PICI reagent gas [19].

For improved quantification, isotope-coded derivatization with trideuterated methylchloroformate (D3-MCF)/D3-methanol medium provides a mixture of internal standards of D3-FAME standards, each corresponding to a fatty acid metabolite in human serum. The mixed FAMEs/D3-FAMEs are eluted nearly together with close retention times and are used to compensate for unacceptable matrix effects and instrument drifts, thus circumventing the matrix calibration problem [24].

The method was validated according to FDA guidelines with linear regressions ≥ 0.98, and the LLOQ ranged from 0.022 to 0.18 µg/mL as well as other validation parameters with satisfactory results for all FA target analytes. GC-MS analysis is the limiting step of FA analysis and is completed unattended in less than 20 min, with sample preparation performed simultaneously in a batch of six serum samples. Overall, the new GC-PICI method enables the analysis of 32 lipid FAs in human serum from 60 samples in 8 h.

Finally, the performance of the method for the study of fatty acid metabolism was evaluated through a quantitative GC-PICI-MS analysis of the SRM 2378 reference human Serum 1, as well as human sera from three local healthy volunteers. The ability of the new method to measure FA profiles in esterified lipids was examined in detail and compared to FA reference data reported for total FA content in two NIST 2378 reference serum materials, each collected from three volunteers with and without fish oil supplementation.

The new GC-PICI-SIM-MS method indeed provided lower FA values compared to the NIST reference data, which is due to the fact that only lipid FAs are measured and not the total content, which also includes free FAs. The summarized data in Table 5 and also in Table 6 show that the total FA content changed by approximately −11.0, −16.8, −23.5 and −20.8% (expressed in fold-change values) for ∑SAFAs, ∑MUFAs, ∑PUFAs (n − 6), and ∑PUFAs (n − 3), respectively. The reduced total content of the most abundant FAs (approximately 17.7%) gives an indirect indication of the level of circulating free FAs in human serum. The decrease in lipid PUFAs (n − 3) was slightly less than that of PUFAs (n − 6) (FC = −20.8 vs. −23.5) compared to the reported total FA content in SRM 2378 Serum3, which is also reflected in the ∑PUFA (n − 6)/(n − 3) ratio (Table 5) with values of 5.2–5.6, indicating a slightly higher ratio of ∑PUFAs (n − 3) to ∑PUFAs (n − 6) bound in serum lipids than in their free form.

The GC-MS examination of three additional serum samples from donors with Central European diets revealed similar FA profiles from individuals with similar diets but different from those in the SRM 2378 reference sera, which were presumably from donors with Western diets in the USA area. The described method enabled a comparison of the measured FA profiles with SRM 2378 reference data and confirmed a considerable lowering of the PUFA (n − 6)/(n − 3) ratio in SRM 2378 Serum 3 and Serum 1 samples after fish oil supplementation to more acceptable anti-inflammatory levels of about 5–5.6 [3,32].

## 5. Conclusions

The described GC-PICI-MS method was validated for the quantification of serum FAs bound in esterified lipids without determining the free FAs present in the biological material. The new method is simple, sufficiently fast, and cost-effective, and it uses inexpensive commercial reagents. The identification and quantification of lipid FAs is facilitated via the addition of easy-to-prepare isotope-coded internal D3-FAME standards, which can be correlated with the majority of FAs detected in the biological sample. By minimizing matrix effects and instrument bias, this approach improves the validation parameters of the FA assay, such as lower LLOQ and quantification range. As in other recent work [19], PICI has been shown to be advantageous over the usual EI mode, as it provides clear, well-defined mass spectra with pseudomolecular [M+H]^+^ ions and provides an option for generating diagnostic ions via MS/MS scanning in tandem MS instruments. Furthermore, the approach can be extended to the exclusive FA analysis in esterified lipids without free FAs and sphingolipids after the application of common lipidomic extraction methods [7] or after the fractionation of lipid classes, and thus, it serves as a complementary method for the characterization of the FA profiles in lipidomic analysis. Finally, the method offers a perspective for the analysis of lipid FAs in dry blood spots and blood cells without interference from free FAs due to the small amount of sample required.

## Figures and Tables

**Figure 1 metabolites-15-00104-f001:**
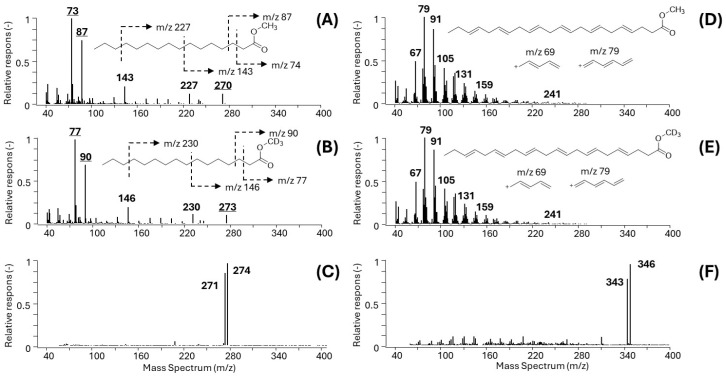
EI and PICI mass spectra of representative unsaturated and polyunsaturated FAMEs. (**A**) EI-MS spectrum of C16:0-Me; (**B**) EI-MS spectrum of the D3-labeled methyl ester (C16:0-MeD3); (**C**) PICI-MS spectrum of GC-MS-coeluting C16:0-Me and C16:0-MeD3 with dominant [M+H]+, *m*/*z* 271 and *m*/*z* 274, respectively; (**D**) EI spectrum of DHA-Me; (**E**) EI spectrum of isotopically labeled DHA-MeD3; (**F**) PICI spectrum of coeluting DHA-Me and DHA-MeD3, [M+H]+ ions, *m*/*z* 343 and *m*/*z* 346, respectively.

**Figure 2 metabolites-15-00104-f002:**
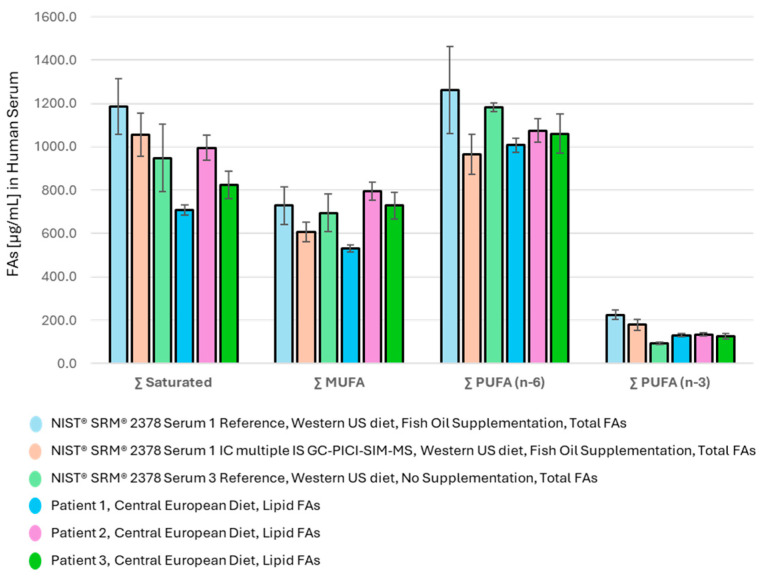
Distribution of major FA families in human serum (saturated, MUFA, PUFA (n − 6), and PUFA (n − 3)); (1) in SRM 2378 Serum 1 (fish oil supplementation), (2) in Serum 3, both with a Western US diet, and (3) in sera of donors on a Central European diet.

**Figure 3 metabolites-15-00104-f003:**
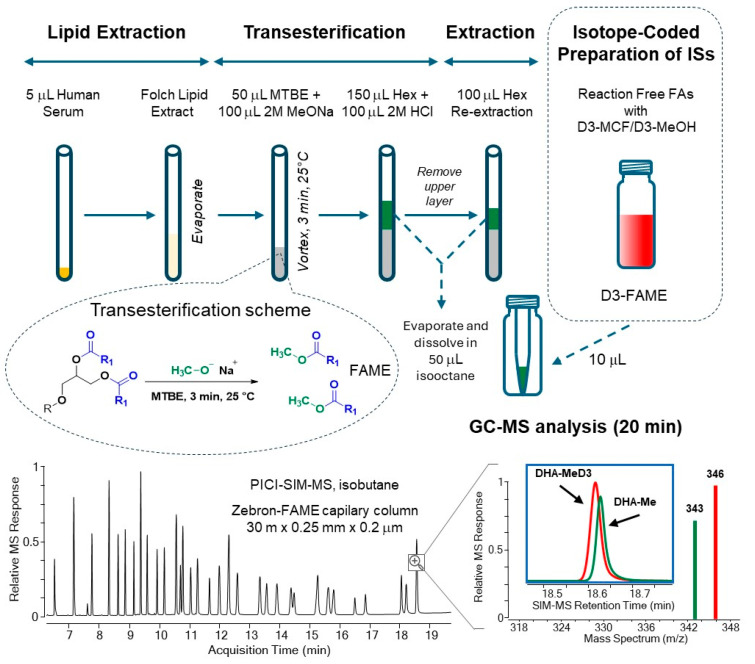
Analytical workflow of the new validated GC-PICI-SIM-MS method for the quantification of 32 lipid fatty acids as FAMEs in human serum via base-catalyzed transmethylation and isotope-coded preparation of D3-FAME internal standards.

**Table 1 metabolites-15-00104-t001:** Recovery of the tested lipid classes during methoxide-catalyzed transmethylation in MTBE medium (n = 6).

Lipids and FFAs	Abbreviation	Recovery [%]	RSD [%]
1-hexadecanoyl-rac-glycerol	MG 16:0	98.4	3.3
1,2-dihexadecanoyl-rac-glycerol	DG 16:0/16:0	90.5	9.4
1,2,3-trihexadecanoyl-sn-glycerol	TG 16:0/16:0/16:0	87.6	5.3
1,2-diheptadecanoyl-sn-glycero-3-phosphate	PA 17:0/17:0	92.2	2.3
1,2-ditetradecanoyl-sn-glycero-3-phosphocholine	PC 14:0/14:0	93.9	3.8
1,2-dihexadecanoyl-sn-glycero-3-phosphoethanolamine	PE 16:0/16:0	100.3	3.0
1,2-di-(9Z-octadecenoyl)-sn-glycero-3-phosphoethanolamine	PE 18:1/18:1	82.2	5.1
1,2-di-(9Z-octadecenoyl)-sn-glycero-3-phospho-(1′-sn-glycerol)	PG 18:1/18:1	97.5	4.2
1,2-ditetradecanoyl-sn-glycero-3-phosphoserine	PSer 14:0/14:0	83.9	2.8
1-(9Z-octadecenoyl)-sn-glycero-3-phosphate	LysoPA 18:1	81.2	4.9
2-tetradecanoyl-sn-glycero-3-phosphoethanolamine	LysoPE 14:0	85.5	5.0
1-octadecanoyl-sn-glycero-3-phospho-(1′-sn-glycerol)	LysoPG 18:0	92.2	3.4
1-(9Z-heptadecenoyl)-glycero-3-phosphoserine	LysoPSer 17:1	83.4	8.3
Cholest-5-en-3β-yl octadecenoate	CHE 18:0	83.8	2.4
O-hexadecanoyl-R-carnitine	Carnitine 16:0	94.2	5.9
*N*-(heptadecanoyl)-sphing-4-enine-1-phosphocholine	SM 17:0	<0.1	-
*N*-(dodecanoyl)-sphing-4-enine	Ceramide 12:0	<0.3	-
*N*-(dodecanoyl)-1-β-glucosyl-sphing-4-enine	GluCer 12:0	<0.4	-
Pentadecanoic acid	C15:0	<0.1	-
Eicosanoic acid	C20:0	<0.1	-
9E-octadecenoic acid	C18:1(n − 9)t	<0.1	-
9Z,12Z-octadecadienoic acid	C18:2(n − 6)	<0.1	-

The lipids are annotated according to the Lipid Maps system [10].

**Table 2 metabolites-15-00104-t002:** Diagnostic ions for the GC-SIM-MS detection of 32 FAMEs via the EI and PICI modes.

		Diagnostic Ions							
Analyte	RT	EI (Analytes)			EI (IS)			PICI (Analytes)	PICI (IS)
		*m*/*z* (1)	*m*/*z* (2)	*m*/*z* (3)	*m*/*z* (1)	*m*/*z* (2)	*m*/*z* (3)	*m*/*z*	*m*/*z*
C8:0	4.31	158	74	87	161	77	90	159	162
C10:0	5.84	186	74	87	189	77	90	187	190
C12:0	7.18	214	74	87	217	77	90	215	218
C13:0	7.78	228	74	87	231	77	90	229	232
C14:0	8.36	242	74	87	245	77	90	242	245
C14:1(n − 5)	8.66	240	74	87	243	77	90	241	244
C15:0	8.90	256	74	87	259	77	90	257	260
C16:0	9.43	270	74	87	273	77	90	271	274
C16:1(n − 7)	9.64	268	74	87	271	77	90	269	272
C17:0	9.96	284	74	87	287	77	90	285	288
C17:1(n − 7)	10.21	282	74	87	285	77	90	283	286
C18:0	10.64	298	74	87	301	77	90	299	302
C18:1(n − 9)t	10.73	296	74	87	299	77	90	297	300
C18:1(n − 9)	10.77	296	74	87	299	77	90	297	300
C18:2(n − 6)t	11.34	294	87	59	297	90	62	295	298
C18:2(n − 6)	11.35	294	87		297	90		295	298
C18:3(n − 6)	11.73	292	194	87	295	197	90	293	296
C18:3(n − 3)	12.10	292	87	236	295	90	239	293	296
C20:0	12.45	326	74	87	329	77	90	327	330
C20:1(n − 9)	12.71	324	74	87	327	77	90	325	328
C20:2(n − 6)	13.47	322	87		325	90		323	326
C21:0	13.69	340	74	87	343	77	90	341	344
C20:3(n − 6)	14.05	320	87	74	323	90	77	321	324
C20:4(n − 6)	14.52	318			321			319	322
C20:3(n − 3)	14.61	320			323			321	324
C22:0	15.49	354	74	87	357	77	90	355	358
C22:1(n − 9)	15.86	352	74	87	355	77	90	353	356
C20:5(n − 3)	15.96	316			319			317	320
C23:0	16.99	368	74	87	371	77	90	369	372
C24:0	18.19	382	74	87	385	77	90	383	386
C24:1(n − 9)	18.38	380	87	74	383	90	77	381	284
C22:6(n − 3)	18.72	342			345			343	346

**Table 3 metabolites-15-00104-t003:** Comparison of the range of quantification [µg/mL] and RSDs [%] of the investigated calibration methods for each FAME analyte.

Method	A. Single-IS EI-MS		B. IC-Multi-IS EI-MS		C. Single-IS PICI-MS		D. IC-Multi-IS PICI-MS	
FAME	Ranges of Quantification	LLOQ RSD [%]	Ranges of Quantification	LLOQ RSD [%]	Ranges of Quantification	LLOQ RSD [%]	Ranges of Quantification	LLOQ RSD [%]
C8:0	0.183–23.5	3.37	0.183–93.7	1.91	0.091–93.4	5.01	0.046–93.4	2.95
C10:0	0.180–23.5	1.64	0.180–92.4	1.41	0.090–92.4	4.35	0.045–92.4	1.52
C12:0	0.180–23.5	3.03	0.180–2.3	1.81	0.180–93.6	7.14	0.180–93.6	2.91
C13:0	0.046–23.3	6.88	0.046–11.7	3.19	0.023–46.7	1.97	0.023–46.7	0.90
C14:0	0.183–23.2	1.99	0.365–12.2	1.84	0.183–93.5	3.69	0.046–93.5	1.43
C14:1(n − 5)	0.182–23.3	0.53	0.046–5.8	3.94	0.046–46.7	2.09	0.023–46.7	2.74
C15:0	0.023–23.2	13.93	0.045–11.6	2.24	0.023–23.2	2.24	0.045–46.5	2.31
C16:0	0.273–35.5	4.23	0.273–35.0	1.21	0.068–70.1	0.69	0.137–140.1	0.79
C16:1(n − 7)	0.047–47.7	7.57	0.047–11.9	2.83	0.047–24.4	2.36	0.023–47.7	4.97
C17:0	0.044–22.4	0.21	0.084–11.2	2.05	0.022–22.4	6.41	0.044–44.8	1.10
C17:1(n − 7)	0.047–24.0	3.09	0.093–12.0	1.43	0.093–47.9	3.26	0.047–47.9	3.94
C18:0	0.092–93.9	2.76	0.183–93.9	0.78	0.046–47.5	2.07	0.046–93.9	1.73
C18:1(n − 9)t	0.022–44.7	8.79	NA		0.022–22.3	2.02	0.022–44.7	1.78
C18:1(n − 9)	0.092–47.5	1.13	0.183–11.7	4.53	0,092–93.9	0.91	0.046–93.9	5.43
C18:2(n − 6)t	0.180–45.6	6.68	0.022–11.4	7.26	0.022–23.5	1.70	0.022–45.6	4.74
C18:2(n − 6)	0.180–47.3	6.00	0.023–11.8	3.97	0.023–24.3	4.68	0.023–47.3	3.69
C18:3(n − 6)	0.180–23.5	1.74	0.183–11.7	5.45	0.023–23.5	1.22	0.023–46.9	0.60
C18:3(n − 3)	0.092–47.0	4.40	0.023–11.8	7.16	0.023–24.5	2.57	0.023–47.0	2.08
C20:0	0.186–94.5	0.75	0.186–47.3	1.43	0.046–47.3	3.77	0.046–94.5	3.20
C20:1(n − 9)	0.180–46.8	0.43	0.091–11.7	2.35	0.091–23.4	2.19	0.047–46.8	0.48
C20:2(n − 6)	1.460–46.8	3.04	0.182–11.7	7.46	0.046–46.8	5.83	0.023–46.8	6.71
C21:0	0.180–46.8	0.78	0.040–11.7	8.84	0.023–23.4	3.36	0.023–46.8	2.93
C20:3(n − 6)	0.047–24.1	6.74	NA		0.047–24.1	3.72	0.023–48.2	7.87
C20:4(n − 6)	0.045–23.2	1.51	NA		0.725–23.2	2.89	0.023–46.4	7.76
C20:3(n − 3)	0.087–45.0	5.25	0.174–22.4	12.33	0.699–22.4	1.45	0.044–22.4	7.90
C22:0	0.182–93.3	0.99	0.046–23.3	8.58	0.090–47.1	2.32	0.046–93.3	8.32
C22:1(n − 9)	0.370–47.1	1.05	NA		0.046–23.7	2.87	0.046–47.1	15.00
C20:5(n − 3)	0.084–21.5	10.41	NA		0.042–21.5	2.39	0.042–21.5	9.42
C23:0	0.170–11.1	0.58	0.021–11.1	15.37	0.087–44.4	9.01	0.087–11.1	10.00
C24:0	0.370–94.1	1.26	NA		0.092–94.1	2.92	0.046–94.1	5.54
C24:1(n − 9)	1.470–46.9	5.66	0.046–11.7	5.16	0.180–46.9	5.49	0.023–46.9	5.69
C22:6(n − 3)	0.092–47.4	0.64	NA	1.91	0.023–24.3	6.88	0.046–47.4	12.65
Average		3.7		4.6		3.4		4.7

NA = values not available due to interfering fragment ions in the coeluting FAME and its D3-FAME pairs in the recorded EI spectra. The lowest value of the range is equal to the LLOQ value.

**Table 4 metabolites-15-00104-t004:** (**A**). Validation parameters of the single-IS GC-EI-SIM-MS method. (**B**). Validation parameters of the IC-multi-IS GC-SIM-PICI-MS method.

(**A**)											
**FAME**	**Calibration Curve**		**Precision (CV [%])**				**Accuracy [%]**				**Ranges of Quantification**
	**Regression Line**	**R^2^**	**LLOQ**	**Low** **QC**	**Medium** **QC**	**High** **QC**	**LLOQ**	**Low** **QC**	**Medium** **QC**	**High** **QC**	**[µg/mL]**
C8:0	y = 0.8648x + 0.0073	0.9983	3.37	4.22	2.13	0.99	103	127	106	88	0.183–23.5
C10:0	y = 0.8864x − 0.0053	0.9905	1.64	1.21	1.85	13.36	81	103	100	89	0.180–23.5
C12:0	y = 1.2164x − 0.0018	0.9995	3.03	0.19	0.10	0.03	92	118	122	85	0.180–23.5
C13:0	y = 1.5193x − 0.0033	0.9981	6.88	2.43	1.04	0.34	89	92	107	90	0.046–23.3
C14:0	y = 1.1725x − 0.0070	0.9996	1.99	1.34	0.65	0.43	121	106	101	87	0.183–23.2
C14:1(n − 5)	y = 0.1785x − 0.0030	0.9991	0.53	1.33	0.35	0.40	91	97	103	93	0.182–23.3
C15:0	y = 1.1309x − 0.0046	0.9988	13.93	2.66	0.83	0.00	104	94	104	94	0.023–23.2
C16:0	y = 1.2985x − 0.0083	0.9997	4.23	2.03	0.79	0.32	88	109	117	88	0.273–35.5
C16:1(n − 7)	y = 0.6513x − 0.0035	0.9991	7.57	2.37	0.95	0.21	86	98	102	94	0.047–47.7
C17:0	y = 1.6466x − 0.0099	0.9972	0.21	3.56	0.34	1.13	80	80	97	97	0.044–22.4
C17:1(n − 7)	y = 1.0234x − 0.0084	0.9982	3.09	3.52	0.18	0.36	115	105	96	96	0.047–24.0
C18:0	y = 1.7571x − 0.1444	0.9985	2.76	3.39	1.11	1.13	101	85	99	97	0.092–93.9
C18:1(n − 9)t	y = 0.5991x − 0.0085	0.9973	8.79	7.21	0.03	1.28	90	96	106	93	0.022–44.7
C18:1(n − 9)	y = 2.4115x − 0.0092	0.9995	1.13	9.73	0.49	0.26	105	76	89	102	0.092–47.5
C18:2(n − 6)t	y = 0.6864x − 0.0148	0.9971	6.68	0.81	2.54	0.47	103	85	94	99	0.180–45.6
C18:2(n − 6)	y = 0.5903x − 0.0108	0.9985	6.00	6.84	0.74	0.84	106	80	89	100	0.180–47.3
C18:3(n − 6)	y = 0.2870x − 0.0067	0.9992	1.74	2.39	0.19	0.94	101	92	100	96	0.180–23.5
C18:3(n − 3)	y = 0.2874x − 0.0016	0.9982	4.40	2.02	5.13	2.52	116	210	76	102	0.092–47.0
C20:0	y = 2.1778x − 0.0403	0.9976	0.75	1.37	0.44	2.84	102	86	93	99	0.186–94.5
C20:1(n − 9)	y = 1.3961x − 0.0244	0.9967	0.43	8.54	1.61	2.44	110	88	91	101	0.180–46.8
C20:2(n − 6)	y = 0.5982x − 0.0358	0.9972	3.04	1.54	0.15	1.54	87	90	100	100	1.460–46.8
C21:0	y = 2.3365x − 0.0556	0.9946	0.78	3.02	0.71	2.90	116	83	87	101	0.180–46.8
C20:3(n − 6)	y = 2.6294x − 0.0194	0.9930	6.74	1.94	0.35	1.87	111	81	91	95	0.047–24.1
C20:4(n − 6)	y = 2.5061x − 0.0164	0.9974	1.51	9.17	14.44	13.50	82	73	115	83	0.045–23.2
C20:3(n − 3)	y = 3.3064x − 0.0268	0.9970	5.25	3.93	1.22	6.29	85	70	89	99	0.087–45.0
C22:0	y = 2.4066x − 0.0387	0.9958	0.99	2.60	1.46	0.24	97	77	91	100	0.182–93.3
C22:1(n − 9)	y = 1.5481x − 0.0566	0.9935	1.05	1.50	2.03	2.73	103	81	86	102	0.370–47.1
C20:5(n − 3)	y = 3.2896x − 0.0422	0.9992	10.41	4.77	1.09	1.88	102	74	102	90	0.084–21.5
C23:0	y = 2.4795x − 0.0678	0.9973	0.58	3.05	2.59	1.82	102	70	89	109	0.170–11.1
C24:0	y = 2.3837x − 0.1272	0.9869	1.26	2.30	0.65	0.21	122	75	84	99	0.370–94.1
C24:1(n − 9)	y = 1.6454x − 0.1225	0.9891	5.66	3.81	2.80	4.49	77	78	93	103	1.470–46.9
C22:6(n − 3)	y = 2.7694x − 0.0237	0.9973	0.64	4.02	1.29	1.99	86	56	77	101	0.092–47.4
(**B**)											
**FAME**	**Calibration Curve**		**Precision (RSD [%])**				**Accuracy [%]**				**Ranges of Quantification**
	**Regression Line**	**R^2^**	**LLOQ**	**Low** **QC**	**Medium** **QC**	**High** **QC**	**LLOQ**	**Low** **QC**	**Medium** **QC**	**High** **QC**	**[µg/mL]**
C8:0	y = 0.044421x + 0.006586	0.9991	2.95	3.36	2.95	2.99	112	96	94	99	0.046–93.4
C10:0	y = 0.034702x + 0.006729	0.9991	1.52	3.36	2.87	2.25	105	99	98	98	0.045–92.4
C12:0	y = 0.082740x + 0.011580	0.9989	2.91	2.63	2.85	1.62	86	80	98	97	0.180–93.6
C13:0	y = 0.067952x + 0.008966	0.9993	0.90	1.20	2.85	1.78	106	99	97	98	0.023–46.7
C14:0	y = 0.035838x + 0.014784	0.9990	1.43	2.11	2.81	1.19	82	94	100	97	0.046–93.5
C14:1(n − 5)	y = 0.062693x + 0.007128	0.9990	2.74	2.51	2.85	1.56	115	100	98	97	0.023–46.7
C15:0	y = 0.084334x + 0.017704	0.9987	2.31	1.81	2.95	1.11	89	99	94	95	0.045–46.5
C16:0	y = 0.047038x + 0.0436814	0.9978	0.79	1.57	2.97	1.27	82	98	105	94	0.137–140.1
C16:1(n − 7)	y = 0.038496x + 0.004194	0.9992	4.97	3.37	2.94	1.20	105	97	97	98	0.023–47.7
C17:0	y = 0.177612x + 0.058213	0.9967	1.10	1.79	3.14	0.87	85	99	106	92	0.044–44.8
C17:1(n − 7)	y = 0.048395x + 0.005794	0.9987	3.94	4.69	3.04	1.16	120	104	99	96	0.047–47.9
C18:0	y = 0.033797x + 0.005894	0.9974	1.73	2.63	2.91	0.80	91	97	97	96	0.046–93.9
C18:1(n − 9)t	y = 0.107772x + 0.021827	0.9883	1.78	3.14	3.01	10.15	87	98	101	98	0.022–44.7
C18:1(n − 9)	y = 0.072445x + 0.000584	0.9955	5.43	3.38	2.99	6.68	84	102	99	96	0.046–93.9
C18:2(n − 6)t	y = 0.033203x + 0.002258	0.9989	4.74	2.63	2.90	1.38	108	97	94	98	0.022–45.6
C18:2(n − 6)	y = 0.032044x + 0.002257	0.9989	3.69	1.93	2.90	1.38	106	98	94	98	0.023–47.3
C18:3(n − 6)	y = 0.146237x + 0.152019	0.9957	0.60	1.23	1.95	0.94	100	108	100	94	0.023–46.9
C18:3(n − 3)	y = 0.024105x + 0.002638	0.9993	2.08	2.80	2.92	1.93	120	97	93	99	0.023–47.0
C20:0	y = 0.043131x + 0.006331	0.9947	3.20	2.50	3.07	0.21	90	95	99	94	0.046–94.5
C20:1(n − 9)	y = 0.053086x + 0.007816	0.9956	0.48	1.96	3.06	0.79	122	98	98	95	0.047–46.8
C20:2(n − 6)	y = 0.035234x + 0.001038	0.9986	6.71	5.64	3.25	0.80	113	98	93	98	0.023–46.8
C21:0	y = 0.086359x + 0.017468	0.9933	2.93	3.46	2.95	0.31	105	102	96	94	0.023–46.8
C20:3(n − 6)	y = 0.42075x + 0.000673	0.9986	7.87	4.27	3.24	1.39	110	98	94	98	0.023–48.2
C20:4(n − 6)	y = 0.046109x + 0.002527	0.9985	7.76	2.49	1.50	0.07	115	115	87	98	0.023–46.4
C20:3(n − 3)	y = 0.200742x + 0.001243	0.9964	7.90	3.93	3.96	0.65	86	100	108	103	0.044–22.4
C22:0	y = 0.66042x + 0.002884	0.9929	8.32	4.76	3.33	1.14	87	89	102	90	0.046–93.3
C22:1(n − 9)	y = 0.027032x + 0.001728	0.9974	15.00	3.80	1.81	0.88	104	85	85	101	0.046–47.1
C20:5(n − 3)	y = 0.04466x + 0.002449	0.9983	9.42	6.15	2.68	0.14	109	99	94	94	0.042–21.5
C23:0	y = 0.246018x + 0.005347	0.9939	10.00	8.00	3.49	1.98	85	113	114	85	0.087–11.1
C24:0	y = 0.063509x + 0.003896	0.9842	5.54	12.80	0.54	0.26	108	85	95	108	0.046–94.1
C24:1(n − 9)	y = 0.032070x + 0.004694	0.9827	5.69	8.21	4.35	15.00	104	87	99	98	0.023–46.9
C22:6(n − 3)	y = 0.008756x + 0.001292	0.9987	12.65	4.37	2.82	2.32	95	88	90	100	0.046–47.4

**Table 5 metabolites-15-00104-t005:** Results of the quantification of lipid FAMEs via two GC-MS methods in NIST^®^ SRM^®^ 2378 serum.

Lipid FA	Single IS GC-EI-SIM-MS		IC Multiple IS GC-PICI-SIM-MS			Reference Serum	
	Concentration ± SD (n = 6)	CV	Concentration ± SD (n = 6)	CV	*p*-Value *	NIST^®^ SRM^®^ 2378 Serum 1 **	Fold Change Measured vs. Reference ***
	[µg/mL]	%	[µg/mL]	%	-	[µg/mL]	%
C10:0	1.92 ± 0.96	50.1	2.69 ± 0.29	10.6	0.135	3.64 ± 0.03	−25.9
C12:0	9.48 ± 1.11	11.7	14.45 ± 3.4	23.5	-	-	-
C14:0	46.4 ± 3.4	7.3	40.8 ± 4.6	11.4	0.059	45.7 ± 1.6	−10.6
C14:1(n − 5)	3.62 ± 0.14	3.9	2.94 ± 0.22	7.5	0.00029	3.62 ± 1.58	−19.0
C15:0	6.46 ± 0.58	9.0	5.92 ± 0.37	6.2	0.115	5.26 ± 0.17	12.6
C16:0	743 ± 56	7.6	730 ± 71	9.7	0.757	851 ± 90	−14.2
C16:1(n − 7)	51.2 ± 4.9	9.5	47.6 ± 3.9	8.2	0.223	54.4 ± 3.8	−12.6
C17:0	7.16 ± 0.33	4.6	6.97 ± 0.44	6.2	0.445	7.28 ± 0.49	−4.3
C18:0	198 ± 7	3.4	220 ± 19	8.8	0.052	226 ± 26	−2.8
C18:1(n − 9)	494 ± 21	4.2	525 ± 40	7.7	0.168	619 ± 68	−15.2
C18:1(n − 7)	NQ	-	NQ	-	-	12 ± 2	-
C18:2(n − 6)	749 ± 66	8.8	769 ± 61	7.9	0.622	1049 ± 179	−26.7
C18:3(n − 6)	9.2 ± 0.5	5.5	11.9 ± 0.6	5.4	3.65 E-05	12.6 ± 1.7	−5.2
C18:3(n − 3)	27.4 ± 1.1	4.1	26 ± 2	7.6	0.216	33.1 ± 4.2	−21.4
C20:0	1.96 ± 0.31	15.8	2.99 ± 0.25	8.5	0.00022	7.81 ± 1.13	−61.8
C20:1(n − 9)	3.97 ± 0.67	16.8	5.82 ± 0.34	5.8	0.00072	6.12 ± 1.02	−4.8
C20:2(n − 6)	4.4 ± 0.5	12.0	4.6 ± 0.3	6.8	0.310	-	-
C20:3(n − 6)	ND	-	27.9 ± 5	18.0	-	-	-
C20:4(n − 6)	144 ± 15	10.5	152 ± 26	17.2	0.540	201 ± 21	−24.1
C20:3(n − 3)	ND	-	ND	-	-	-	-
C22:0	12.3 ± 0.3	2.4	15.3 ± 0.2	1.3	2.72 E-08	19.4 ± 4.4	−21.2
C22:1(n − 9)	ND	-	2.31 ± 0.27	11.8	-	1.69 ± 1.02	36.2
C20:5(n − 3)	73.1 ± 5.1	7.0	71.9 ± 3.7	5.1	0.678	85.9 ± 11.2	−16.3
C24:0	21.9 ± 4.8	22.0	16.6 ± 0.1	0.4	0.0588	19.9 ± 4.8	−16.5
C24:1(n − 9)	21.8 ± 0.4	1.8	23.4 ± 0.3	1.2	4.67 E-05	32.6 ± 9.5	−28.4
C22:6(n − 3)	77.4 ± 15.6	20.1	80.4 ± 19.5	24.2	0.788	106.1 ± 5.3	−24.2
∑SAFA	1049 ± 75	7.1	1056 ± 100	9.4		1187 ± 128	−11.0
∑ MUFA	574 ± 27	4.7	607 ± 45	7.4		729 ± 87	−16.8
∑PUFA (n − 6)	906 ± 82	9.1	966 ± 93	9.6		1262 ± 202	−23.5
∑PUFA (n − 3)	178 ± 22	12.3	178 ± 25	14.1		225 ± 21	−20.8
∑ FAs	2602 ± 196	7.5	2691 ± 255	9.4		3271 ± 410	−17.7
∑PUFA (n − 6)/(n − 3)	5.1		5.2			5.6	
∑PUFA(n − 3)/∑FAs	5.8%		5.7%			5.9%	

CV = coefficient of variation; SD = standard deviation; ND = not detected; NQ = not quantified; MM= relative molecular mass; SAFA = saturated FAs; MUFA = monounsaturated FAs; PUFA = polyunsaturated FAs; ∑FAs = concentration sum of all quantified FAs. * *p*-value calculated using Welch’s *t*-Test. ** = The original NIST Serum 1 values in [µmol/L] were recalculated to [µg/mL] = tabulated concentration in the NIST Serum 1 [µmol/L] * MM(FA)/1000. *** The fold-change (FC) value here describes the change between the FA concentration measured with the IC multi-IS GC-PICI-SIM-MS method and the tabulated reference value NIST^®^ SRM^®^ 2378. The FC value is calculated as follows: FC = (measured value−reference value)/reference value*100). ∑PUFA (n − 6)/(n − 3) is calculated as the ratio of the sum concentrations of linoleic acid (C18:2(n − 6)), gamma-linolenic acid (C18:3(n − 6)), and arachidonic acid (C20:4(n − 6)) to the sum of alpha-linolenic acid (C18:3(n − 3)), eicosapentaenoic acid (C20:5(n − 3)), and docosahexaenoic acid (C22:6(n − 3)). ∑PUFA (n − 3)/∑ FAs is calculated here as the percentage of eicosapentaenoic acid (EPA, C20:5(n − 3)) and docosahexaenoic acid (DHA, C22:6(n − 3)) in relation to the quantified most abundant FAs in the measured and SRM^®^ 2378 Serum 1 samples (i.e., C14:0, C16:0, C16:1(n − 7), C18:0, cC18:1(n − 9), cC18:2(n − 6c), cC18:3(n − 3), C20:3(n − 6), C20:4(n − 6), C20:3(n − 3), C20:5(n − 3), and cC22:6(n − 3)).

**Table 6 metabolites-15-00104-t006:** Results of the quantification of lipid FAMEs using the IC-multi-IS GC-PICI-MS method in three patients and comparison with the reference values in NIST^®^ SRM^®^ 2378 Serum 3.

Lipid FA	Patient 1 Serum		Patient 2 Serum		Patient 3 Serum		Average FA Serum Levels in 3 Patients	Reference Serum	
FAME	Concentration ± SD (n = 6)	CV	Concentration ± SD (n = 6)	CV	Concentration ± SD (n = 6)	CV	Concentration ± SD	NIST^®^ SRM^®^ 2378 Serum 3 *	Fold Change Measured vs. Reference **
	[µg/mL]	[%]	[µg/mL]	[%]	[µg/mL]	[%]	[µg/mL]	[µg/mL]	[%]
C10:0	0.28 ± 0.031	11.0	0.2 ± 0.038	18.6	0.06 ± 0.003	5.9	0.18 ± 0.024	0.91 ± 0.16	−80.1
C12:0	0.92 ± 0.36	39.2	1.02 ± 0.41	40.3	1.2 ± 0.48	40.2	1.05 ± 0.21	-	-
C14:0	20.7 ± 1.19	5.8	34.9 ± 3.78	10.8	40.3 ± 2.95	7.3	32 ± 2.64	35.4 ± 0.9	−9.7
C14:1(n − 5)	1.69 ± 0.06	3.3	2.33 ± 0.16	7.0	3.25 ± 0.2	6.3	2.42 ± 0.14	3.17 ± 1.81	−23.6
C15:0	6.67 ± 0.25	3.8	6.04 ± 0.47	7.8	8.62 ± 0.64	7.4	7.11 ± 0.46	5.02 ± 0.12	41.7
C16:0	507 ± 16.41	3.2	699 ± 39.45	5.6	586 ± 44.73	7.6	597 ± 33.53	656 ± 118	−9.0
C16:1(n − 7)	40.7 ± 1.44	3.5	59.1 ± 3.84	6.5	76.9 ± 6.67	8.7	58.9 ± 3.99	46.8 ± 3.3	25.8
C17:0	5.64 ± 0.21	3.8	6.3 ± 0.47	7.4	6.33 ± 0.56	8.8	6.09 ± 0.41	7.17 ± 0.38	−15.0
C18:0	153 ± 5.84	3.8	234 ± 13.99	6.0	169 ± 15.43	9.2	185 ± 11.75	198 ± 21	−6.4
C18:1(n − 9)	483 ± 15.8	3.3	723 ± 37.04	5.1	642 ± 53.61	8.4	616 ± 35.49	582 ± 68	5.8
C18:1(n − 7)	NQ	-	NQ	-	NQ	-		33 ± 3	-
C18:2(n − 6)	836 ± 26.78	3.2	865 ± 45.08	5.2	859 ± 75.84	8.8	853 ± 49.24	934 ± 6	−8.6
C18:3(n − 6)	5.4 ± 0.17	3.1	8.1 ± 0.45	5.6	10.5 ± 0.92	8.8	8 ± 0.51	14.9 ± 1	−46.5
C18:3(n − 3)	24.6 ± 1.21	4.9	41.4 ± 2.95	7.1	24.4 ± 2.67	10.9	30.1 ± 2.28	17.4 ± 0.1	73.4
C20:0	4.94 ± 0.03	0.5	5.19 ± 0.05	1.0	4.87 ± 0.04	0.8	5 ± 0.04	8.13 ± 2.81	−38.5
C20:1(n − 9)	4.39 ± 0.17	3.9	8.66 ± 0.53	6.1	4.56 ± 0.25	5.4	5.87 ± 0.31	6.02 ± 0.43	−2.5
C20:2(n − 6)	4.33 ± 0.17	3.8	9.57 ± 0.54	5.7	5.88 ± 0.46	7.9	6.59 ± 0.39	-	-
C20:3(n − 6)	23.5 ± 1.22	5.2	50.9 ± 2.66	5.2	49 ± 4.02	8.2	41.1 ± 2.63	-	-
C20:4(n − 6)	138 ± 4.77	3.4	140 ± 5.2	3.7	136 ± 9.12	6.7	138 ± 6.36	233 ± 14	−40.7
C20:3(n − 3)	22.3 ± 1.22	5.5	23.7 ± 1.46	6.2	22.5 ± 2.36	10.5	22.9 ± 1.68	-	-
C22:0	4 ± 0.06	1.6	4.1 ± 0.03	0.7	3.9 ± 0.03	0.8	4 ± 0.04	19.8 ± 19.8	−79.8
C22:1(n − 9)	0.2 ± 0.02	9.1	2.07 ± 0.21	10.3	1.85 ± 0.07	4.0	1.37 ± 0.1	2.03 ± 2.03	−32.3
C20:5(n − 3)	33.5 ± 1.69	5.0	24.7 ± 1.31	5.3	28.4 ± 2.47	8.7	28.8 ± 1.82	19.3 ± 19.3	49.5
C24:0	4.93 ± 0.022	0.4	4.97 ± 0.015	0.3	4.91 ± 0.015	0.3	4.94 ± 0.017	18.1 ± 18.1	−72.7
C24:1(n − 9)	0.89 ± 0.12	14.0	1.21 ± 0.09	7.3	0.93 ± 0.11	11.8	1.01 ± 0.11	22.4 ± 22.4	−95.5
C22:6(n − 3)	50 ± 3.13	6.3	45 ± 2.22	4.9	51 ± 4.27	8.4	49 ± 3.21	56.2 ± 56.2	−13.3
∑SAFA	708 ± 24	3.4	995 ± 59	5.9	824 ± 65	7.9	843 ± 49	949 ± 154	−11.2
∑MUFA	531 ± 18	3.3	796 ± 42	5.3	729 ± 61	8.4	685 ± 40	695 ± 87	−1.4
∑PUFA (n − 6)	1007 ± 33	3.3	1074 ± 54	5.0	1060 ± 90	8.5	1047 ± 59	1182 ± 21	−11.4
∑PUFA (n − 3)	136 ± 7	5.4	141 ± 8	5.7	132 ± 12	9.0	136 ± 9	93 ± 5	47.0
∑FAs	2333 ± 81		2941 ± 159		2683 ± 224		2652 ± 155	2778 ± 236	−4.5
∑PUFA (n − 6)/(n − 3)	9.1		9.1		9.7		9.3	12.7	
∑PUFA(n − 3)/∑FAs	3.6%		2.4%		3.0%		3.0%	2.7%	

CV = variation coefficient; SD = standard deviation; ND = not detected; NQ = not quantified; MM = relative molecular mass. The age and mass body index (BMI) of healthy patients 1, 2, and 3 were 42, 41, and 64 years, 23.7, 27.7, and 27.8, respectively. * The original NIST Serum 3 values in [µmol/L] were recalculated to [µg/mL] = tabulated concentration in the NIST Serum 3 [µmol/L] * MM(FA)/1000. ** The fold-change (FC) value here describes the change between the FA concentration measured with the IC multi-IS GC-PICI-SIM-MS method and the tabulated reference value NIST^®^ SRM^®^ 2378, Serum 3. The FC value is calculated as follows: FC = (measured value−reference value)/reference value*100). PUFA (n − 6)/(n − 3) is calculated as the ratio of the sum concentrations of linoleic acid (C18:2(n − 6)), gamma-linolenic acid (C18:3(n − 6)), and arachidonic acid (C20:4(n − 6)) to the sum of alpha-linolenic acid (C18:3(n − 3)), eicosapentaenoic acid (C20:5(n − 3)), and docosahexaenoic acid (C22:6(n − 3)). ∑ PUFA (n − 3)/∑ FAs is calculated here as the percentage of eicosapentaenoic acid (EPA, C20:5(n − 3)), and docosahexaenoic acid (DHA, C22:6(n − 3)) in relation to the quantified most abundant FAs in the measured and SRM^®^ 2378 Serum 1 samples (i.e., C14:0, C16:0, C16:1(n − 7), C18:0, cC18:1(n − 9), cC18:2(n − 6c), cC18:3(n − 3), C20:3(n − 6), C20:4(n − 6), C20:3(n − 3), C20:5(n − 3), and cC22:6(n − 3)).

**Table 7 metabolites-15-00104-t007:** Overview of advantages and disadvantages of the tested calibration methods.

Calibration Method	Pros	Cons
(A) Single-IS GC-EI-SIM-MS	Acceptable statistics in quantitation range	Highest LLOQ Narrower quantitation range Diagnostic ions in PUFA of low intensity
(B) IC-multi-IS GC-EI-SIM-MS	Acceptable statistics in quantitation range Wider dynamic range	Higher LLOQ for most FAs Diagnostic ions in PUFA of low intensity PUFA quantitation not available Preparation of ISs
(C) Single-IS GC-PICI-SIM-MS	Diagnostic [M+H]+ for all FAs Wide quantitation range	Higher LLOQ Narrower quantitation range
(D) IC-multi-IS GC-PICI-SIM-MS	Diagnostic [M+H]+ for all FAs Lowest LLOQ Widest quantitation range (mostly over three orders)	Preparation of ISs

## Data Availability

The data presented in this study are available upon request from the corresponding authors.

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
