# Peer review of "A New, Validated GC-PICI-MS Method for the Quantification of 32 Lipid Fatty Acids via Base-Catalyzed Transmethylation and the Isotope-Coded Derivatization of Internal Standards"

_metabolites, 2025, doi:10.3390/metabo15020104_

Round 1

Reviewer 1 Report

Comments and Suggestions for Authors

This study proposed GC-PICI-MS method for the simple, cheap ,and efficient quantification of serum FAs. This study is well-established and only needs a minor revision before publication.

1. Key words need to be updated to reflect the key point of this study. 

2. Line 52: The introduction of CG-MS here needs more background.

3. Line 71: although ->though

4. Line 270: The relevant data can be shown in supporting materials.

5. Line 344: It is recommended to use a table to summarize the advantages/disadvantages of these four calibration methods.

Author Response

First, we would like to thank the Reviewer 1 for his/her time and effort in reviewing the submitted work and requesting minor revision.

  1. Key words need to be updated to reflect the key point of this study.

RE: Thank you for this warning. The missing term “fatty acid analysis“ has been added.

  1. Line 52: The introduction of GC-MS here needs more background.

RE: Thank you for this comment pointing out a missing reference in line 52. We have provided an original reference [16] in the context of the FA analysis in lines 52-55. The other references to the GC-MS background are presented in lines 70-103 of the Introduction section, References [7-26]. The Endnote system re-numbered the references accordingly. We have checked the references independently. 

  1. Line 71: although ->though

RE: Thank you for this comment. We can change the term to the though. Nevertheless, the use of both terms can be considered.  

  1. Line 270: The relevant data can be shown in supporting materials.

RE: Thank you for this recommendation. We have clarified the text in line 270. However, in our opinion, further discussion of transesterification yields would already distract from the main results of the paper, which focuses on the quantification of FAs.

  1. Line 344: It is recommended to use a table to summarize the advantages/disadvantages of these four calibration methods.

Re: Thank you for this recommendation. For the sake of clarity, we have therefore inserted Table 7 in line 503 of the Discussion.

Reviewer 2 Report

Comments and Suggestions for Authors

This work by Vodrážka and co-workers describes a sound and robust approach for fatty acid analysis that follows FDA guidelines and requires very low volumes of serum.

The study introduces a GC-MS method validated according to FDA guidelines, quantifies a broad range of 32 fatty acids, focusing on esterified lipids, it adds isotope-coded internal standards to reduce matrix effects and bias, and is versatile to be easily applied. The manuscript is easy to read, and contains all the necessary information.

I don't find any flaws in the work; the chemistry of FA esterification can be tricky, but that's organic chemistry. I have 1 comments on the work and 1 other not on the work itself but on its importance/relevance, which I think the authors downplayed a bit. Other than that, congratulations are in order.

1. Line 106 - “Supelco 37 Component FAME Mix” contains 37 fatty acids (I checked this description: https://www.sigmaaldrich.com/PT/en/product/supelco/crm47885?context=product), and you mention 32, which I believe are the 32 in Table S2 (actually it would be good to point this out). why did you skip 5? For example, the butyrate and hexanoate might be interesting when working with plants.

2. One aspect that needs to be covered is the improvement brought upon by this method. This should be clearly discussed to enhance the impact of the research. Some possible questions are:

- What are the advantages of this method over, for example, GC-TOF based approaches? - Or LC-MS approaches such as lipidomics? How does this method compare to other methods (e.g., LC-MS, GC-TOF) in terms of quantitative accuracy, especially in complex biological matrices? Does the GC-PICI-MS method achieve lower limits of detection/quantification for certain fatty acids compared to LC-MS or GC-TOF?

- What are the advantages and disadvantages of using esterification? Why can't you do without it? I think this is critical. Lipidomics are typically carried out without modifications and relying on extensive databases. How does this compare? Could the separation technique described here outperform lipidomics approaches that rely heavily on databases? For example, how well does this method resolve isomers or double-bond positional differences in fatty acids? How does it perform in terms of sample processing speed and scalability for high-throughput studies, such as clinical trials or population-level research?

- How easy is it to standardize /validate this method for global approaches compared to lipidomics approaches that depend on databases requiring constant updates? Is the whole process easily transferable to other labs?-

- What are  the limitations brought upon by  esterification? do you loose information regarding the lipid structures?

Author Response

First, we would like to thank the Reviewer 2 for his/her time and effort in reviewing the submitted work and requesting a minor revision.

  1. Line 106 - “Supelco 37 Component FAME Mix” contains 37 fatty acids (I checked this description: https://www.sigmaaldrich.com/PT/en/product/supelco/crm47885?context=product), and you mention 32, which I believe are the 32 in Table S2 (actually it would be good to point this out). why did you skip 5? For example, the butyrate and hexanoate might be interesting when working with plants.

RE: Thank you for this thoughtful comment. Indeed, we have only investigated and validated the newly reported method for 32 FAs. The reason for this is simply due to the availability of the internal fatty acid standards and their volatility. In general, the low fatty acids up to the C10 fatty acids are quite volatile and the method would require further specific measures that are beyond the scope of this work. Moreover, we have only reported on the analysis of fatty acids bound in lipids, and low fatty acids are rarely present there and therefore not of particular interest for lipid analysis. Nevertheless, we appreciate this comment and have tried to improve the text in this context in line 303 and also slightly change the heading of Supplementary Table S2, as recommended, for better clarity.

  1. One aspect that needs to be covered is the improvement brought upon by this method. This should be clearly discussed to enhance the impact of the research.

RE: Thank you for the proposed final considerations. We are aware that good scientific work raises more questions than answers. We hope that the debate on the reported results will stimulate future research on this topic.

Our reflections are summarized in the Conclusions. We have emphasized that our work has produced five main messages:

(1) Direct basic transesterification in the medium polar solvent TMBE gives satisfactorily high FA yields from major lipid classes (not free FAs and sphingolipids) in three minutes;

(2) Calibration with FAMEs produced from free FAs by isotope-coded esterification can increase the quantification range and LLOQ of each FA released from lipids;

(3) GC-PICI-MS has been shown to be a valuable method for the quantification of lipid FAs in complex biological matrices;

(4) The proposed method is a complementary method in lipid research for future detailed characterization of FAs bound in specific lipid classes

(5) The method is very promising in GC-PICI-MS analysis of small sample amounts such as dry blood spots and blood cells.

Some possible questions are:

- What are the advantages of this method over, for example, GC-TOF based approaches? - Or LC-MS approaches such as lipidomics? How does this method compare to other methods (e.g., LC-MS, GC-TOF) in terms of quantitative accuracy, especially in complex biological matrices? Does the GC-PICI-MS method achieve lower limits of detection/quantification for certain fatty acids compared to LC-MS or GC-TOF?

RE: As we have also emphasized in the paper, the method described here represents a complementary approach to the “main stream” lipidomic LC-MS methods. The method is fast, inexpensive and can be performed with relatively inexpensive GC-MS instruments. However, its full potential still needs to be investigated in future applications. GC-TOF instruments equipped with an industrial standard PICI probe, for example, make the method directly applicable for numerous applications.

- What are the advantages and disadvantages of using esterification? Why can't you do without it?

I think this is critical. Lipidomics are typically carried out without modifications and relying on extensive databases. How does this compare? Could the separation technique described here outperform lipidomics approaches that rely heavily on databases? For example, how well does this method resolve isomers or double-bond positional differences in fatty acids? How does it perform in terms of sample processing speed and scalability for high-throughput studies, such as clinical trials or population-level research?

RE: As we pointed out in the Conclusions, the method focuses on a small part of the lipidome, on the analysis of FAs bound as acyls (esterified in lipids). Nothing more and nothing less.

The method described does not surpass current lipidomic strategies, but it complements them by providing an independent knowledge of lipid FAs.

- How easy is it to standardize /validate this method for global approaches compared to lipidomics approaches that depend on databases requiring constant updates? Is the whole process easily transferable to other labs?

- What are the limitations brought upon esterification? Do you loose information regarding the lipid structures?

RE: The standardization of the method has been achieved by its full validation according to FDA guidelines. Of course, the method can be easily transferred to any laboratory dealing with FA analysis. Of course, it provides information about the FAs bound in the lipids. For a comprehensive characterization of lipids, other commonly used lipidomic methods are then required, as mentioned in the Reviewer´s comment.

Reviewer 3 Report

Comments and Suggestions for Authors

The authors have extensively investigated and presented their data extensively and excellently. 

I have no hesitation recommending its acceptance as it is. 

Author Response

The authors have extensively investigated and presented their data extensively and excellently. I have no hesitation recommending its acceptance as it is. 

Re: We would like to thank the Reviewer 3 for his/her time and effort in reviewing the submitted work.

Round 2

Reviewer 2 Report

Comments and Suggestions for Authors

I consider the article can be publised in its present form.

Author Response

Comment: I consider the article can be publised in its present form.

Re: Thank you very much for the review and the positive evaluation of the submitted article.